# Investigating Mammalian Formins with SMIFH2 Fifteen Years in: Novel Targets and Unexpected Biology

**DOI:** 10.3390/ijms24109058

**Published:** 2023-05-21

**Authors:** Metello Innocenti

**Affiliations:** Department of Biotechnology and Biosciences, University of Milano-Bicocca, Piazza della Scienza 2, 20126 Milan, Italy; metello.innocenti@unimib.it or metelloinnocenti@gmail.com

**Keywords:** formins, SMIFH2, actin, microtubules, filopodia, cell migration, interferon, myosin, HIV-1, cancer

## Abstract

The mammalian formin family comprises fifteen multi-domain proteins that regulate actin dynamics and microtubules in vitro and in cells. Evolutionarily conserved formin homology (FH) 1 and 2 domains allow formins to locally modulate the cell cytoskeleton. Formins are involved in several developmental and homeostatic processes, as well as human diseases. However, functional redundancy has long hampered studies of individual formins with genetic loss-of-function approaches and prevents the rapid inhibition of formin activities in cells. The discovery of small molecule inhibitor of formin homology 2 domains (SMIFH2) in 2009 was a disruptive change that provided a powerful chemical tool to explore formins’ functions across biological scales. Here, I critically discuss the characterization of SMIFH2 as a pan-formin inhibitor, as well as growing evidence of unexpected off-target effects. By collating the literature and information hidden in public repositories, outstanding controversies and fundamental open questions about the substrates and mechanism of action of SMIFH2 emerge. Whenever possible, I propose explanations for these discrepancies and roadmaps to address the paramount open questions. Furthermore, I suggest that SMIFH2 be reclassified as a multi-target inhibitor for its appealing activities on proteins involved in pathological formin-dependent processes. Notwithstanding all drawbacks and limitations, SMIFH2 will continue to prove useful in studying formins in health and disease in the years to come.

## 1. Introduction

Formin family proteins are key regulators of the cytoskeleton and are involved in a wide variety of essential processes in eukaryotic cells [1]. Different formins are endowed with variable biochemical activities on actin and tubulin, but their functions are typically fulfilled by formin homology (FH) 1 and 2 domains (Figure 1), which form an evolutionarily conserved tandem [1,2,3]. The FH1 domain harbors several profilin-binding motifs and modulates the nucleation and elongation of linear actin filaments with the adjacent FH2 domain, which forms a donut-shaped dimer [1,2]. In mammals, the formin family comprises fifteen proteins [1,2,3]. Diaphanous-related formin 3 (DIAPH3) or its functionally equivalent and 85% identical mouse homologue Diap3 (aka mDia2), its paralogues DIAPH1 and DIAPH2 (mDia1 and mDia3 in mice, respectively), and seven other formins share a common multi-domain architecture and make up the Diaphanous-related formin (Drf) subfamily [4] (Figure 1). The N-terminus of Drfs comprises a GTPase-binding domain (GBD) for activated Rho A-C and F, followed by a Diaphanous Inhibitory Domain (DID) and a Dimerization Domain (DD) that mediates homo-oligomerization [1,2,3]. Interestingly, DIAPH3 also harbors a partial Cdc42 Interactive Binding region (CRIB), to which activated Cdc42 binds, partially embedded in the GBD [5]. The C-terminal region of Drfs has a Diaphanous Auto-regulatory Domain (DAD) that interacts with the N-terminal DID, thereby holding Drfs in a ‘closed’ auto-inhibited conformation [1,2,3]. The binding of activated Rho to the GBD and the phosphorylation of two residues flanking the DAD can break the DID–DAD interaction and convert Drfs to the ‘open’ active conformation [1,2,3]. In keeping with this, mutations disrupting the DID–DAD interaction render Drfs constitutively active [6,7]. Furthermore, the DAD of mDia1 may assist actin nucleation using the FH2 [8], although it is unknown whether this is a general Drf property. 

The FH1-FH2 of the majority of mammalian formins is sufficient to stabilize microtubules (MTs) via acetylation and to induce coalignment between actin stress fibers and MTs in cells [9]. In particular, the FH1-FH2 of mDia2 was shown to bind and to stabilize MTs independently of actin binding and nucleation with the assistance of the C-terminal region [10]. However, the mDia2-mediated actin assembly is required for stable MTs to be able to reach the cell periphery [10], and the binding of MTs to the C-terminal region of mDia2 inhibits actin nucleation [11], thus pointing toward an interplay between the regulation of actin and the MTs. Upon activation by upstream signals and conversion to the open conformation, full-length mDia1 and mDia2 may also stabilize cellular MTs indirectly by forming a complex with MT-regulatory proteins [12].

The unique ability to control both actin and MT dynamics qualifies formins as ideal master regulators of the crosstalk between cytoskeletal networks [1]; accordingly, all eukaryotes rely on one or more formin-family protein(s) to locally modulate the actin cytoskeleton and MTs. Formins have been implicated in several developmental and homeostatic cellular processes such as actin-based mitochondrial fission [13], actin- and MT-based vesicle and organelle trafficking [14,15], actin-based ciliogenesis [16], actin-based membrane protrusion [17,18,19], actin-dependent endocytic pathways [20,21], the establishment and maintenance of polarity for (asymmetric) cell division [22,23], and cell migration [1]. They also affect actin-dependent cell signaling and gene transcription, and genomic integrity [24,25]. Not surprisingly, germline and somatic mutations perturbing formins’ activity and/or expression are associated with a growing number of pathological conditions, including developmental defects of the heart, nervous system, and kidney and aging-related diseases, inherited human diseases, and cancer [26,27].

However, the large number of formins and the seemingly functional redundancy among certain family members make it cumbersome to attribute a given cellular function to a specific formin by means of gene knockout or knockdown approaches. Furthermore, this is also a roadblock to studies requiring rapid inhibition of formin function. In this context, both pan-formin as well as formin-specific inhibitors would greatly benefit research in the field. As such, the discovery of small molecule inhibitor of formin homology 2 domains (SMIFH2) [28] has provided a unique tool to explore formins’ functions from the molecular to the organismal scales. Due to the important pathophysiological roles of formins in eukaryotes, SMIFH2 has been so widely used that a Google Scholar search for ‘SMIFH2’ returned 835 entries as of May 2023. About half of them appear to be peer-reviewed scholarly contributions (listed in Appendix A).

Here, I critically discuss the literature regarding the characterization of SMIFH2 as a pan-formin inhibitor as well as the growing evidence of unexpected off-target effects. Available data are compared and contrasted to highlight outstanding controversies and open questions regarding the substrates and mechanism of action of SMIFH2. Whenever possible, I propose sensible explanations for seemingly discrepant observations and roadmaps to answer key open questions in the field. The review of published studies along with deposited hitherto neglected results open the door to the possible exploitation of SMIFH2 as a dual-specificity inhibitor in the context of HIV-1 infection and of cancer-cell migration.

## 2. SMIFH2 Discovery and Characterization In Vitro and in Cells

Small molecule inhibitor of formin homology 2 domains (SMIFH2) (Figure 2), a 2-thiooxodihydropyrimidine-4,6-dione derivative, was identified in 2009 among approximately 10,000 commercially available small molecules (ChemBridge, San Diego, CA, USA) screened for the ability to prevent actin assembly induced by the FH1-FH2 of mDia1 and of mDia2 in vitro [28].

SMIFH2 was also able to fully inhibit profilin-actin assembly [half maximal inhibitory concentration (IC_50_) 15 μM] and exhibited specificity for formin-mediated actin assembly; indeed, it did not affect spontaneous polymerization of actin monomers or barbed-end elongation, or actin polymerization stimulated by the Arp2/3 complex [28]. This and the fact that it is active on at least six of the seven mammalian formin subfamilies (IC_50_ from 6 to 30 μM, see Appendix A for details) [29] as well as on formins from evolutionarily distant species (IC_50_ from 5 to 15 μM) [28] qualify SMIFH2 as a general inhibitor of actin assembly mediated by formins. The IC_50_ for formin-mediated elongation of actin filaments was 4.0 μM [28], suggesting that SMIFH2 is more potent on formin-induced actin-filament elongation than nucleation. Consistent with formins associating processively with the actin filament’s barbed end and thereby accelerating its elongation in the presence of profilin-actin [1], SMIFH2 decreased the affinity of formins for the barbed ends with an IC_50_ of about 30 μM [28], thus less efficiently than nucleation and elongation. However, it should be noted that the IC_50_ for formin-dependent nucleation obtained from TIRF-based single-filament actin polymerization assays returned a more closely matching value (25 μM) [28].

Treatment of fission yeast cells with 25 μM SMIFH2 for 30 min resulted in the disappearance of formin-dependent actin cables and contractile actin rings [28], whereas actin patches, the formation of which relies on the activity of the Arp2/3 complex, were not affected [28]. Nevertheless, a much lower concentration of SMIFH2 was sufficient to induce a full disassembly of the actin cables as compared with the actin rings (2.5 μM vs. 25 μM) [28]. This could be due to either the inherently different stability of these actin structures or to Fur3 being more sensitive than Cdc12 to SMIFH2. Interestingly, 10 μM SMIFH2 perturbed the localization of type II myosin (Myo2) at the contractile ring, even though this actin-based structure was intact [28]. Despite the fact that these observations hinted that SMIFH2 could interfere with the activity of myosins, as recently proven in animal cells [30], this possibility was neglected.

SMIFH2 induced cytotoxicity in immortalized murine fibroblasts (NIH3T3 cells) (IC_50_ 28 μM at 24 h post treatment) and the appearance of large, presumably apoptotic blebs already at 6 h after the addition of 30 μM SMIFH2 [28]. Such cytotoxic effects might have a cell-type-specific threshold, as a much higher IC_50_ (75 μM) was observed in A539 lung carcinoma cells. This contention should however be taken with caution because 30 μM of SMIFH2 fully arrested cell growth in both cell lines [28]. Consistent with this latter finding and the multinucleation of SMIFH2-treated NIH3T3 cells [28], mitotic events were rare in cells exposed to 25 μM SMIFH2 [31]. In addition, massive rapid cell death that was independent of p53 status or levels was observed in a number of mammalian cell lines at SMIFH2 concentrations higher than 25 μM [31].

Lamellipodium formation in NIH3T3 cells was reduced by SMIFH2 in a dose-dependent manner and fully suppressed upon treatment with 30 μM SMIFH2 for 6 h [28]. Accordingly, 10 μM of SMIFH2 halved the migration speed of NIH3T3 cells and reduced the percentage of cells with thick stress fibers [28]. The SMIFH2-induced loss of stress fibers was dose-dependent, and the 5% of the treated cells devoid of stress fibers showed instead peripheral lamellipodia and reduced focal-adhesion size [28]. Six SMIFH2 analogues were tested, showing that their ability to inhibit formins in vitro could be correlated with the disruption to actin cables and the contractile ring in fission yeast, whereas none of them perturbed the actin patches. Surprisingly, these analogues did not affect the actin cytoskeleton in NIH3T3 cells [28]. A subsequent in-depth systematic study revealed that lamellipodium/ruffle formation increased between three and five hours of SMIFH2 treatment, and this temporally correlated with enhanced migratory abilities [31]. Directionality was not concomitantly affected, suggesting that SMIFH2 affects cell motility by transiently modulating actin-based protrusion [31].

Although formins regulate MT dynamics, whether and how SMIFH2 affects MTs was not investigated until several years later. In mammalian cells, a single dose of 25 μM SMIFH2 was found to induce alternated depolymerization/repolymerization cycles of actin and tubulin [31]. To shed light on this unexpected result, cells were treated with 25 μM SMIFH2, and the SMIFH2-contaning medium was replaced every two hours. Under these conditions, progressive and persistent depolymerization of both F-actin and MTs could be observed. Given that the SMIFH2-containing medium was prepared at the beginning of the time course, the depolymerization–repolymerization cycles of actin and MTs were due to intracellular SMIFH2 breakdown and/or inactivation rather than intrinsic instability [31]. The integrity of the Golgi complex was damaged by SMIFH2, which was in good agreement with its effects on actin and MT dynamics [31].

The mechanism of action of SMIFH2 remains ill-defined, and the binding site on its target proteins are hitherto unknown. Notwithstanding this, the discrepancies outlined above raise the possibility that SMIFH2 and/or its analogues could have species-specific off-target effects and affect the activity of other proteins in addition to inhibiting formins. In recent years, unappreciated SMIFH2 targets have been discovered serendipitously [32] and through both candidate approaches [30] and unbiased high-throughput screens (see PubChem bioassays below).

## 3. Identification of Mammalian Myosins and Interferons as SMIFH2 Targets

### 3.1. Myosins

Formins and myosins often act together in cells because many actin-based structures built and/or regulated by formins are not only decorated with but also remodeled by myosins [1,2,19,33,34]. Starting from the observation that treatment of fibroblasts with 30 μM SMIFH2 inhibited the contraction of stress fibers and the movement of actin arcs, it has recently been discovered that SMIFH2 can inhibit actomyosin contractility and myosin-decorated actin-filament flow in both living and permeabilized cells [30]. SMIFH2 inhibited the ATP-induced actomyosin contractility in a dose-dependent manner (IC_50_ of about 50 μM) in permeabilized cells [30]. The centripetal movement of transverse actin arcs, which could also be induced by the addition of ATP to permeabilized cells, was more sensitive to SMIFH2 than actomyosin contractility, being fully inhibited at 50 μM [30]. The effects of SMIFH2 on stress-fiber retraction and actin-arc movement in permeabilized cells mimicked those of blebbistatin [30], a myosin II inhibitor [35]. Of note, permeabilized cells did not contain G-actin, and the employed ATP-containing buffer was supplemented with phalloidin, which stabilizes actin filaments [30]. As actin dynamics were not possible in this system, the results casted doubts on the selectivity of SMIFH2 for formins.

In vitro, SMIFH2 inhibited the basal ATPase activity of both human skeletal muscle and rabbit non-muscle myosin 2A (IC_50_ around 50 μM), although the maximal inhibition was achieved at concentrations far above 100 μM and plateaued at 73% [30]. Despite incomplete inhibition, the absence of actin in these assays proved that SMIFH2 targets myosin rather than indirectly inhibiting actin binding. SMIFH2 also interfered with the ability of both human skeletal muscle and rabbit non-muscle myosin 2 to translocate actin filaments, as assessed using gliding actin in vitro motility assays [30]. Inhibition of human skeletal muscle myosin 2 was complete at high SMIFH2 concentrations (above 100 μM) and irreversible, whereas rabbit non-muscle myosin 2A, which is partially phosphorylated and thus more active, could only be mildly inhibited by 250 μM SMIFH2 [30]. Although SMIFH2 affected myosin 2A only at high concentrations, the basal ATPase activity of myosins from other classes was inhibited at doses like those used to arrest formin-induced actin polymerization (bovine myosin 10: IC_50_ of about 15 μM, Drosophila myosin 7 and myosin 5: IC_50_ of about 30 and 2 μM, respectively) [30].

Direct inhibition of myosin 2 by SMIFH2 could in principle contribute to the effects of SMIFH2 observed in cells, but three main reasons make this possibility unlikely. Firstly, the concentration of SMIFH2 sufficient to stop non-muscle myosin 2A-dependent processes in cells (30 μM [30]) is far below that needed to achieve the inhibition of its activity in vitro (250 μM [30]). Secondly, the effects of SMIFH2 on non-muscle myosin 2-mediated gliding of actin filaments are irreversible in vitro [30], whereas those observed in cells were both reversible [28,36] and transient [31]. Last but not least, SMIFH2 enhanced mesenchymal-like cell migration on 2D surfaces [31], a phenotype that would hardly be compatible with non-muscle myosin 2 inhibition. Hence, it is more plausible that disrupted formin-actin filament interactions would account for the effects of SMIFH2 on the contractility and flow of myosin-decorated actin filaments [37]. Notwithstanding this point, the similar IC_50_ of SMIFH2 for formins and for myosin 10 would be suitable to effectively counter cancer-cell invasiveness, to which both formin-dependent and myosin 10-dependent filopodia contribute [34,38]. Although being a valuable dual inhibitor, SMIFH2 would not allow us to dissect the molecular dependencies of these filopodia.

### 3.2. Interferons

Dysregulation of Interferon (IFN)-Janus Kinase (JAK)-Signal Transducer and Activator of Transcription protein (STAT) signaling axis contributes to the pathogenesis of autoimmune and inflammatory diseases, as well as cancers [39]. To find new small molecule inhibitors of IFN-induced JAK-STAT signaling, HeLa cells were pre-treated with 40 μM SMIFH2 for 20 min, and the phosphorylation of STAT1 was assessed 20 min after IFNγ stimulation. SMIFH2 virtually abrogated STAT1 phosphorylation and perturbed the actin cytoskeleton, causing the cell area to shrink [32]. However, cell shrinkage could also be due to cytotoxicity, which occurs at SMIFH2 concentrations higher than 25 μM [31]. In any case, SMIFH2 might inhibit the bioactivity of IFNs and/or interfere with JAK-STAT signaling [32], as discussed in Section 5.

Twenty-eight structural variants of SMIFH2 were synthetized to uncouple the effects on JAK-STAT signaling from those of formins and to shed light on the underlying mechanism. Some of them retained the ability to inhibit the IFNγ-induced phosphorylation of STAT1-3 in a dose-dependent manner (from high nM to low μM) without cytotoxic effects at the effective doses, but they no longer affected cell area [32]. Some others showed instead the opposite behavior, suggesting that the inhibition of IFNγ signaling and of formins are independent events [32]. Further experimentation showed that STAT1 activation by EGF was not significantly perturbed by either SMIFH2 or the variants blocking IFN, but not formin, activity, ruling out general non-specific effects [32]. The fact that SMIFH2 blunted type I-III IFN signaling suggests that it might be a pan-IFN inhibitor. However, inhibition of JAK-STAT signaling by SMIFH2 is evident only at concentrations much higher than those currently recommended to avoid major off-target effects (>40 μM vs. <25 μM) [31,32]. Although IFNs are pleiotropic cytokines, SMIFH2 has different potency towards formins and IFNs, and this allows defining experimental conditions that minimize the IFN-dependent effects.

## 4. High-Throughput Bioactivity Assays Reveal Novel SMIFH2 Targets

High-throughput screening (HTS) entails the testing of large numbers of different small molecules (up to a few million), typically in miniaturized, fully automated, and unbiased assays. Since the late 1990s, HTS has been a mainstay of hit identification, the process whereby chemical hits for the development of drug candidates for a certain target are discovered [40]. In recent years, however, the pharmaceutical industry is more and more exploiting drug repurposing, a strategy that employs several approaches to identify new uses for approved or investigational drugs that differ from their original medical indication. As such, drug repurposing is advantageous over the classical ex novo development of a new drug to treat a certain medical condition; it reduces both the risks of failure and the costs and time required to bring a drug to market [41]. Given the involvement of formins in several human pathologies [26], SMIFH2 has been included in many chemical libraries containing repurposable bioactive compounds that are used for drug discovery. An account of bioactivity assays deposited on PubChem in which SMIFH2 inhibited the designated drug target (Table 1) is collated below, along with a critical assessment of any published evidence linking formins with the tested drug target.

### 4.1. Ribonuclease H (RNase H) Activity of the HIV-1 Reverse Transcriptase

Human immunodeficiency virus (HIV) reverse transcriptase (RT) is endowed with three enzymatic activities: an RNA-dependent polymerase activity, a DNA-dependent polymerase activity, and a ribonuclease H (RNase H) activity that is crucial for the degradation of the viral genomic RNA template during first-strand DNA synthesis [42]. These three activities jointly allow RT to convert the single-strand genomic RNA of HIV into a double-strand (ds)DNA, which can be integrated into the host cell’s nuclear DNA. Therefore, RT is essential for the replication of HIV, the causative agent of acquired immunodeficiency syndrome (AIDS), thus representing an appealing anti-HIV druggable target. Indeed, several anti-HIV therapeutics targeting RT are in clinical use, all of which inhibit the DNA polymerase activity. Nevertheless, the increasing prevalence of variants resistant to currently administered medications has spearheaded the quest for compounds that interfere with other stages of HIV replication. For this reason, RT RNase H remains an attractive drug target.

A primary screening for inhibitors of the ribonuclease H (RNase H) activity of the HIV-1 reverse transcriptase (RT) p66/p51 heterodimer was performed in the framework of the Molecular Targets Development Program (MTDP) (PubChem BioAssay AID 372). The screen employed a cell-free enzymatic assay measuring RNase H activity on a fluorescent substrate and was optimized to avoid false-positive hits [43]. Ten thousand substances (ChemBridge) were tested, and SMIFH2 was the eighth most effective compound among the 770 active ones. Ten μM of SMIFH2 inhibited the activity of RNase H by 98% and was assigned a PubChem activity score of ninety-eight based on three replicates. Under these conditions, SMIFH2 is thus a very robust RNase H inhibitor.

HIV-1 hijacks the host-cell cytoskeleton for uncoating and trafficking towards the nucleus of the viral core upon entry into the cytoplasm [44,45]. In more detail, HIV-1 exploits actin filaments for short-range movement near the cell periphery and MTs for long-range bidirectional movement towards the nucleus. Incoming HIV-1 cores appear to hop from the cortical actin cytoskeleton to MTs in a process that relies on several host-cell and viral proteins. On the one hand, MTs and their associating motor proteins favor the HIV-1-infection by promoting transport of the virus and capsid uncoating [45]. On the other hand, HIV-1 can stimulate the formation of stable MTs to optimize its own early infection phase [46]. Interestingly, host-cell proteins that bind to or crosslink either actin filaments or MTs have also been involved in regulating HIV-1 uncoating, reverse transcription, and trafficking along MTs. Among them, mDia1/DIAPH1 and mDia3/DIAPH2 are co-opted by HIV-1 to orchestrate core uncoating and transport [47]. These two formins may coordinate discrete processes during early HIV-1 infection by affecting MT stability and the disassembly of the HIV-1 core, a conical structure composed of approximately 1500 molecules of capsid protein (CA) [44]. The knockdown of DIAPH1 or DIAPH2 strongly reduced HIV-1 entry, an effect mimicked by SMIFH2, HIV-1-induced MT stabilization, and HIV-1 reverse transcription during early stages of infection [47]. Conversely, the infection of non-human retroviruses such as Simian Immunodeficiency Virus and Murine Leukemia Virus was independent of them and did not result in MT stabilization [47], showing the functional specificity of DIAPH1 or DIAPH2 in HIV-1 infection. Overexpression of full-length mDia1 or of a constitutively active mDia2 deletion mutant unable to bind and regulate actin [10] enhanced MT stability and HIV-1 reverse transcription as compared to control cells. Hence, mDia1/DIAPH1 and mDia3/DIAPH2, and possibly also mDia2/DIAPH3, act before or at the initiation of reverse transcription in a way that is independent of their actin-regulatory functions [47]. This might also hold for other family members, given that the ability to induce MT acetylation (i.e., stable MTs) is a general feature of formins [9]. Furthermore, the MT-based retrograde trafficking of the HIV-1 core to the nucleus was impaired upon silencing of mDia1/DIAPH1 or of mDia3/DIAPH2, whereas MT-based mitochondrial movement was not affected [47]. This latter observation is revealing because mDia2/DIAPH3 is key for proper mitochondrial positioning and activity in both normal and cancer-associated fibroblasts [14]. Hence, it is not straightforward to imagine that a common molecular mechanism would underpin the roles of the mDia proteins in HIV-1 infection. Whatever the case, the depletion of mDia1/DIAPH1 or mDia3/DIAPH2 and the overexpression of mDia1 or mDia2 reduced and increased HIV-1 uncoating, respectively [47]. Surprisingly, this could not be explained by MT stabilization, as two parts of the FH2 domain (amino acid 801–910 and 910–1040) were each sufficient to bind assembled HIV-1 CA and to promote its uncoating, but, at odds with a previous study [10], none of them stabilized the MTs [47]. These results suggest that the mDia formins have separable and actin-independent roles in HIV-1 uncoating and in MT stabilization and HIV-1-core transport. Hence, more work is required to clarify whether MT stabilization and HIV-1-core transport truly rely on distinct mDia functional domains. Yet, it appears that HIV-1 does not hijack the actin-regulatory functions of mDia formins, but that it takes advantage of their binding to the viral capsid to coordinate the transition of incoming HIV-1 cores from cortical actin filaments to MTs and MT-based centripetal trafficking with uncoating (Figure 3a).

Some unrelated observations hint at a plausible molecular mechanism; entry of human herpes virus 8 (HHV-8, also known as Kaposi’s sarcoma-associated herpesvirus) into the target cell induces a transient reorganization of MTs into bundles and induces MT stabilization, in addition to increasing the interaction between Rho and mDia3/DIAPH2, which mediates MT stabilization downstream of RhoA [48]. Furthermore, RhoA and mDia3/DIAPH2 were shown to be required for the MT-dependent trafficking of HHV-8 capsids towards the nucleus [48], where viral DNA replication occurs. These results and the fact that Rho and Cdc42 activities are required for optimal HIV-1 infection and retro-transcription [49,50] jointly suggest a model whereby the open conformation of mDia formins allows the FH2 domain to interact with the CA of HIV-1 and MTs in a way that outcompetes or is incompatible with actin binding (Figure 3b). Although this model would agree with the notion that actin-capping protein promotes the mDia1-dependent formation of stable MTs by inhibiting mDia1 translocation to the growing end of actin filaments [51], MT stabilization by mDia2 is likely be conformation-insensitive [52], and even mDia1 might function similarly [51]. Therefore, it is equally possible that mDia formins could couple HIV-1 capsid disassembly with HIV-1-induced MT stabilization and HIV-1-core trafficking to the nucleus using a mechanism that is independent of both the actin-regulatory functions and conformation (Figure 3b). This scenario would reveal an unexpected parallel with p53 regulation by mDia2, which is independent of the actin-regulatory functions, conformation-insensitive, and mediated by the C-terminal region [2,31].

Interestingly, the implication of mDia formins in HIV-1 infection goes beyond the processes discussed above. In infected dendritic cells (DCs), viral egress entails the formation of dynamic HIV-induced filopodia bearing immature virions at their tip, and mDia3/DIAPH2 turned out to be essential for the protrusion of such filopodia [53] (Figure 4).

Furthermore, the filopodia projected by DCs engaged and then tethered the plasma membrane of neighboring T cells and, at those contact sites, budded off viral particles that could infect the bound T cells [53]. Similarly, T-cell nanotubes, a type of tunnelling nanotubes that creates cytoplasmic continuity between T cells [54,55], allow for the rapid intercellular spread of HIV-1 [54]. Considering that T-cell nanotubes derive from filopodia and thus are very likely to be formin-dependent [17,34,56,57,58], this HIV-1 transmission mechanism could be targetable with SMIFH2.

Taking all this into account, it is conceivable that crystal structures of SMIFH2 in complex with the isolated HIV-1 RT RNase H domain or with the FH2 domain of formins could provide strong structural foundations for developing and optimizing SMIFH2-derived dual RT RNase H and formin inhibitors that may efficiently counter HIV-1 infection and intercellular spread. The rise of multi-kinase and multi-target anti-cancer drugs that are currently in use or being evaluated in clinical trials [59] makes such an outlook particularly intriguing.

### 4.2. Dual-Specificity Phosphatase 3 and Dual-Specificity Phosphatase 6

Dual-specificity phosphatases (DUSPs) or Vaccinia-H1-like (VH1-Like) enzymes encompass 63 members with diverse substrate specificity, representing the largest group of class I cysteine-based phosphatases [60]. Among them are 11 typical DUSPs or MAPK specific phosphatases (MKPs), specific for the MAPKs ERK, JNK, and p38. There is also a second group, the VH1-like phosphatases or atypical DUSPs (A-DUSPs), consisting of 20 small and poorly characterized enzymes [60].

DUSP3 phosphatase/Vaccinia H1 Related (VHR) phosphatase, the founding member of the A-DUSP group, comprises 185 amino acids making up a catalytic domain that accepts both phospho-tyrosine and phospho-threonine as substrates [61]. DUSP3 inhibits the activity of both ERK and JNK in several cell lines and functions in vivo as either a tumor suppressor or an oncogene, depending on the cancer type. Furthermore, this enzyme may be involved in immune responses, thrombosis, hemostasis, angiogenesis, and genomic stability. Notwithstanding this, Dusp3 knockout mice are healthy and with no overt phenotype [62], even though DUSP3 affected b-FGF-induced endothelial cell sprouting required for pathophysiological neovascularization [62].

DUSP6 is instead a typical DUSP with a high selectivity for ERK1 and ERK2 due to high-affinity binding between its N-terminal MAPK-binding domain and ERK1/2, which in turn induces a conformational change that activates the phosphatase domain [60]. DUSP6 is constitutively expressed in several immune cell types and is involved in immunity and infection [60]. Yet, Dusp6 knockout mice are healthy and fertile, even though they have increased phospho-ERK levels in the heart, spleen, kidney, and brain [63].

The multi-faceted biological functions of DUSP3/VHR and DUSP6/MPK-3 would make small-molecule inhibitors highly desirable for possible therapeutic applications as well as for accelerating the basic research on these enzymes.

Pubchem bioassay (AID 1992) is a primary screening that belongs to the larger assay project “Summary of the absorbance assay for the identification of compounds that inhibit VHR1” of the Burnham Center for Chemical Genomics. A set of 50,000 drug-like molecules (ChemBridge) was screened in vitro using a colorimetric phosphatase assay containing recombinant DUSP3/VHR, a substrate, and a compound at a working concentration of 0.15 mg/mL. All compounds showing more than 60% inhibition were rescreened at 20 μM, and those exhibiting at least 60% inhibition also in the confirmatory screen (221) were regarded as true hits. SMIFH2 reproducibly inhibited DUSP3/VHR phosphatase activity by 88%. Although these results need to be further validated in cells, it is possible that the SMIFH2 concentrations (below 25 μM, [31]) currently recommended for cell treatment might reduce DUSP3/VHR activity.

DUSP6/MPK-3 activity was measured in a similar way using a fluorescence assay and two compound libraries (ChemBridge and NIH collections) in Pubchem bioassay AID 425. In the primary screening, the compounds of the former collection were screened at 3.333 μg/mL, whereas those belonging to the latter were instead screened in mM. All compounds inhibiting DUSP6/MKP-3 by at least 50% were defined as active and retested at 20 μM. If still active, some of them were further assessed at the dose-response confirmation stage. SMIFH2 completely inhibited MKP-3 activity both in the primary screen and at 20 μM. Hence, SMIFH2 might interfere with DUSP6 activity at the concentrations routinely used to study formins in mammalian cells [31].

Until evidence implicating formins in the regulation of these two phosphatases is provided, the inhibition of DUSP3 and DUSP6 in cells should be viewed as a potential undesirable off-target effect of SMIFH2.

### 4.3. Tyrosine-Protein Phosphatase Non-Receptor Type 7 Isoform 2

Tyrosine-protein phosphatase non-receptor type 7 isoform 2 (PTPN7, also called HePTP) is a tyrosine phosphatase expressed in hematopoietic cells that controls ERK and p38 [64,65]. PTPN7/HePTP is often upregulated in myelodysplastic syndrome, T cell acute lymphoblastic leukemia, and acute myelogenous leukemia [65,66]. Small molecule inhibitors of HePTP would thus be useful to study signal transduction and MAPK regulation and may also have an impact on the treatment of hematopoietic malignancies.

The Burnham Center for Chemical Genomics carried out a screen (bioassay record AID 521) to identify PTPN7/HePTP inhibitors among the compounds of the two libraries described in the section above [66]. This screen was designed as outlined for DUSP6 and relied on a colorimetric assay. All compounds showing at least a 50% inhibition of PTPN7/HePTP in the primary screen were defined as active and retested at 20 μM. SMIFH2 was among the 728 active compounds identified in the primary screen and fully inhibited the activity of PTPN7/HePTP both in the primary and confirmatory screens.

Given that PTPN7/HePTP is the only pTyr-specific PTP known to dephosphorylate ERK and p38 in hematopoietic cells [66] and that formins have important regulatory roles in T cells [67,68,69], SMIFH2-induced phenotypes in these cell types should be interpreted with caution.

### 4.4. Heat Shock 70kDa Protein 1A

Overexpression of molecular chaperones is common in many cancers and protects malignant cells from a wide variety of endogenous and exogenous stressors, including chemotherapeutic agents. The human genome contains nine Hsp70-family genes; *HSPA1A* encodes Heat shock 70 kDa protein 1A (HSPA1A/Hsp70-1), the expression of which is upregulated in diverse types of tumor cells [70]. In conjunction with other heat shock proteins, HSPA1A stabilizes existing proteins against aggregation and mediates the folding of newly translated proteins as well as the refolding or proteasome-dependent degradation of misfolded client proteins both in the cytosol and on organelles [70].

Bioassay record AID 583, performed at the Burnham Center for Chemical Genomics, employed the two libraries named in the previous section to identify compounds that displace fluorescein-labelled ATP from recombinant HSPA1A/Hsp70-1. SMIFH2 was one of the 82 active compounds that were found and, when retested at 20 μM, it almost fully displaced ATP from HSPA1A/Hsp70-1.

Non-cytotoxic SMIFH2 concentrations induce the proteasome-independent degradation of selected proteins, including p53, p300 and mDia2 in cancer and immortalized cells, seemingly acting post-transcriptionally [31]. Interestingly, while the transient or stable silencing of mDia2 or DIAPH3 failed to mimic the downregulation of p53 induced by SMIFH2, reducing the expression of FMN1, a formin that is upregulated upon DIAPH3 silencing [71], or of FBXO3, an adaptor protein to which mDia2, FMN1, and p53 can bind [71], was sufficient to decrease p53 expression at a post-transcriptional level [71]. Noteworthy, mDia2 silencing leads to the upregulation of FMN2 in some cancer cell types and in primary fibroblasts [71,72]. These observations collectively suggest that mDia2/DIAPH3 and FMN proteins share the same p53-regulatory functions [71,72]. How these formins regulate gene expression post-transcriptionally is not yet known.

Given that HSPA1A binds to and regulates the levels of p53, and also specifically interacts with full-length constitutively active mDia2 M1041A [71,73], one could speculate that the SMIFH2-induced destabilization of mDia2 and p53 [14,31,71,72,73] would be a consequence of HSPA1A inhibition. However, this hypothesis has three major caveats. Firstly, SMIFH2 does not induce the proteasomal degradation of mDia2 and p53, nor does it decrease their mRNA levels [31]. Secondly, mDia2 levels are insensitive to proteasome inhibition in many cell lines [31,71,73]. Thirdly, the effects of HSPA1A on p53 levels depend on p53 conformation and on the activity of HSP90 [74], and they involve the ubiquitin proteasome system [70].

The sum of these considerations suggests that if SMIFH2 inhibited HSPA1A/Hsp70-1 in cells, this would simply be an off-target effect. Yet, a recently discovered small molecule inhibitor that induces the proteasome-dependent degradation of DNAJA1, in turn promoting that of conformational mutant p53, also reduced filopodium formation and cell migration [75]. These results leave the door open for a functional link between formins and HSPA1A/Hsp70-1, which might make SMIFH2 useful as a dual inhibitor blocking pathological processes dependent on both activities.

## 5. Possible Mechanism(s) for Formin Inhibition by SMIFH2

### 5.1. Non-Covalent Reversible Inhibition of the FH2 Domain

Three clues hint at SMIFH2 targeting the FH2 domain of formins. Firstly, SMIFH2 inhibited both mDia1’s FH1-FH2 activity (with and without profilin) and that of the FH2 alone, which cannot bind to profilin [28]. Secondly, pre-treatment with SMIFH2 and subsequent washout only partly rescued the actin nucleation activity of formins [28]. Thirdly, the effects of SMIFH2 on NIH3T3 cells were claimed to be reversed within 10 min of washout [28]. Together, these data support the idea that SMIFH2 binds the FH2 domain in a non-covalent reversible manner. A subsequent study showed that mammalian cells exposed to 25 μM of SMIFH2 reacquire a normal cytoskeleton and Golgi complex after sixteen hours of treatment; the study also showed that intracellular SMIFH2 decays within 4–5 h [31]. These data corroborate the notion that SMIFH2 act as a reversible inhibitor [31].

Six analogues (compound **2**–**7**) were compared with SMIFH2 (compound **1**) in the original publication [28]; these analyses suggested that the thiourea moiety is necessary for formin binding, as its substitution with a urea moiety caused a substantial loss in activity (compare compound **1** and **2** with compound **3** [28]) (Figure 2). Surprisingly, a recent study reported that the same substitution did not alter the inhibitory activity towards formins (compound **616l**, [32]). However, this conclusion relied on a single and indirect readout, such as cell area [32].

### 5.2. Covalent Inhibition of Formins

In sharp contrast with the above results, SMIFH2 could bind covalently to some of its biological targets [32]. Most SMIFH2 analogues containing a terminal alkyne on the furan ring became selective inhibitors of IFNγ-induced STAT1 phosphorylation. These analogues formed covalent adducts with the nucleophile groups on IFNγ, possibly including lysines, which do not allow IFNγ to interact with its receptor [32]. Although mass spectrometry analyses failed to identify such adducts, most likely due to the reversible nature of the 1,4 Michael additions and their stability under those experimental conditions, SMIFH2 analogues that were covalently bound to IFNγ could be visualized by using native gels [32]. Preincubation of IFNs with SMIFH2 or its IFN-specific derivatives blocked IFN signaling [32]. This and the marked electrophilic properties of SMIFH2, which make it potentially promiscuous [76], lead to speculation that SMIFH2 might have the same mechanism of action even without those main reactive groups.

In line with the notion that SMIFH2 may covalently modify its targets, SMIFH2 abolished the myosin-mediated translocation of actin filaments [30] and formin-induced actin nucleation [28] in vitro, effects that were not or only partially reversed by extensive washout, respectively. Taken together, these data raise the intriguing possibility that SMIFH2 could covalently bind to and thereby inhibit formins. This awaits validation in vitro and, if substantiated, would also need to be confirmed under more physiologically relevant conditions. How SMIFH2 would covalently modify its targets among the plethora of extracellular and intracellular proteins with similar chemo-physical properties, and why such proteins do not outcompete genuine SMIFH2 targets are the main drawbacks of this model.

## 6. Can SMIFH2 Be Modified to Obtain an Isoform-Specific Formin Inhibitor?

A recent structure–activity study of SMIFH2 against a panel of human formins that relied on in vitro actin polymerization assays [29] and NMR analyses has addressed this question. The testing of a panel of SMIFH2 derivatives showed that many alterations disrupted SMIFH2 activity, whereas some of them increased potency by up to fivefold, and a few others produced SMIFH2 derivatives with mild selectivity for certain formins [29]. However, no isoform-specific formin inhibitor could be identified, although the SMIFH2 chemical space was not fully explored [29].

Interestingly, freshly dissolved SMIFH2 turned out to be an uneven mixture of two isomers, which reached a 1:1 stoichiometry after incubation at room temperature for 20 h [29]. It is unknown whether both isomers or only one of them inhibits actin polymerization induced by formins.

At any rate, the activity data of SMIFH2 and of its 17 analogues indicated that the thiocarbonyl and the furan ring are essential for inhibiting formin activity (Figure 2). These conclusions agree with and extend those drawn using a similar in vitro approach [28], whereas they partially differ from a previous study in which formin inhibition was inferred from cellular phenotypes [32]. Although achieving specificity among highly structurally related domains such as the FH2 could be difficult using the SMIFH2 scaffold, an inhibitor specific for mDia1 and mDia2, but not mDia3, has previously been found, even though its low water solubility has hampered usability [77].

## 7. Conclusions and Future Directions

Looking at SMIFH2 fifteen years after its discovery offers a unique perspective on formins—one showing that great strides have been made in the field thanks to this small molecule inhibitor. It is also evident that SMIFH2 can no longer simply be regarded as a pan-formin inhibitor, and that its usage comes with pros and cons. Six years after the identification of SMIFH2 [28], a peer-reviewed publication reported for the first time possible off-target effects of this inhibitor [31]. Since then, evidence that SMIFH2 targets other proteins in addition to formins has steadily grown. Researchers now know that, under certain treatment regimens, SMIFH2 could inhibit myosins, IFNs, and JAK-STAT signaling [30,31,32]. Furthermore, the on-target effects of SMIFH2 have brought to light the unexpected role of mDia2/DIAPH3 and FMN formins in p53 regulation [2,31,71,72].

Less known to the academics are the results of large screening campaigns, which are often deposited in public repositories. By mining PubChem, one such open repository, additional SMIFH2 targets have been unearthed (Table 1) and discussed herein.

Some of the proven and alleged off-target effects of SMIFH2, particularly the inhibition of proteins involved in pathological formin-dependent processes, might turn out to be beneficial and prompt a reassessment of SMIFH2 as a multi-target inhibitor for the development of new treatments for human diseases. Prime examples are unmet medical needs such as cancers and viral infections.

Despite the widespread use of SMIFH2, two key features of this compound remain obscure. Firstly, it is not yet established whether SMIFH2 inhibits all formins expressed in a given cell type. The reported lability of SMIFH2 in cells [31] and the fact that it induces dynamic cytoskeletal rearrangements [31] and exhibits variable IC_50_ for formins in vitro [28,29] are warning signs. Measuring the intracellular concertation and isomerization [29] of SMIFH2 would help lift some of these concerns. Secondly, the characteristics of the protein–SMIFH2 interactions remain elusive. No structural data are yet available for SMIFH2 bound to the FH2 domain of formins or to myosins, hindering cogent approaches to the design of structural perturbations that improve or alter SMIFH2 specificity. Therefore, an outstanding question is the one as to the structural similarity among SMIFH2-binding sites of different targets. Common structural features could explain the dual (or multi) specificity of SMIFH2, but such similarities could not exist because the electrophilic nature of SMIFH2 allows for promiscuous interactions [76].

Regardless of these pitfalls, there is no doubt that SMIFH2 continues to be valuable for exploring formin (patho)biology. Its ability to rapidly inhibit multiple formins in a large variety of eukaryotic cells make SMIFH2 an unsurpassed chemical tool in the hands of basic and preclinical researchers. For example, SMIFH2 overcomes the functional redundancy among certain formin-family members, which undermines loss-of-function approaches targeting a single formin when establishing formins’ involvement in a certain process. It also remains instrumental for studies requiring the rapid and/or reversible inhibition of formins in cells. As outline herein, off-targets effects should always be carefully assessed prior to drawing conclusions based on SMIFH2 treatment. Of note, there is an optimistic outlook for SMIFH2 because just as the number of off-target effects increased, so too did our ability to find experimental settings that allow for distinguishing between on-target and off-target effects, even in complex biological systems.

The development of novel and more specific pan-formin inhibitors as well as that of formin-specific inhibitors would greatly benefit studies on formins from the molecular to the organismal scale. Considering the important pathophysiological roles of formins in eukaryotes, this represents a timely and highly relevant future research avenue. Notwithstanding all drawbacks and limitations discussed in this review, there is no doubt that SMIFH2 will continue to prove useful in studying formins in health and disease in years to come.

## Figures and Tables

**Figure 1 ijms-24-09058-f001:**
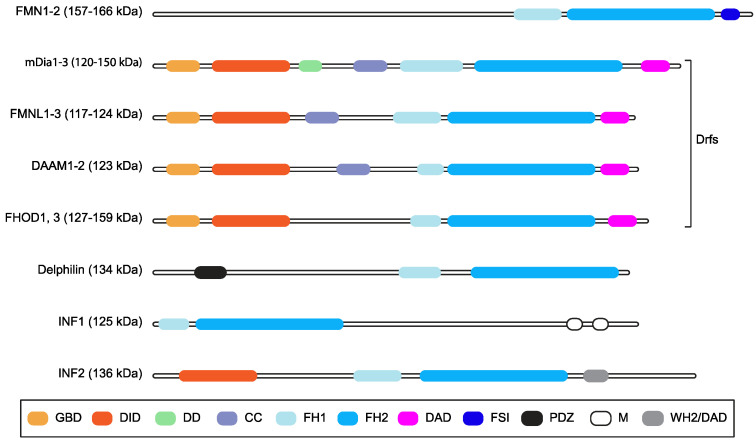
Mammalian formins. Schematic representation of the multi-domain organization of the fifteen mammalian formins, grouping together those belonging to the Drf subfamily. Domains are color-coded, and their names are shown in the box (GBD: GTPase-binding domain; DID: Diaphanous inhibitory domain; DD: dimerization domain, CC: coiled-coil region; FH1: Formin Homology domain 1; FH2: Formin Homology domain 2; DAD: Diaphanous auto-regulatory domain; FSI: Formin-Spire interaction domain; PDZ: Postsynaptic density 95, Discs large, Zona occludens-1 domain; M: microtubule binding domain; WH2/DAD: WASP homology 2-like domain/DAD). Predicted molecular weight or weight ranges are indicated (kDa) close to the subfamily name. The CRIB domain is not displayed because it overlaps with the GBD.

**Figure 2 ijms-24-09058-f002:**
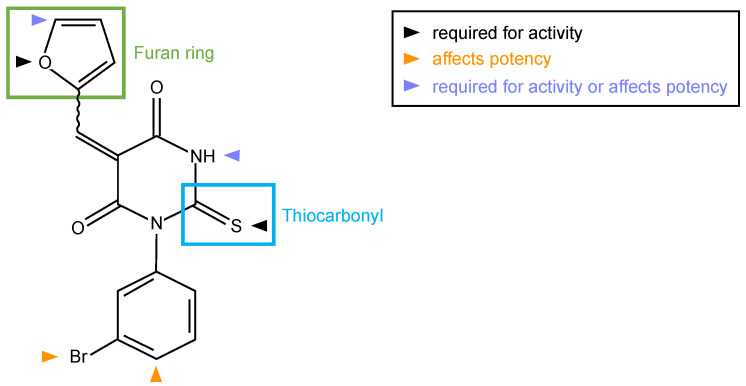
Molecular structure of SMIFH2. Green and cyan boxes mark the furan ring and thiocarbonyl moieties of the inhibitor, respectively. Colored arrowheads indicate positions where modifications influence the activity and/or potency of SMIFH2 analogues [29].

**Figure 3 ijms-24-09058-f003:**
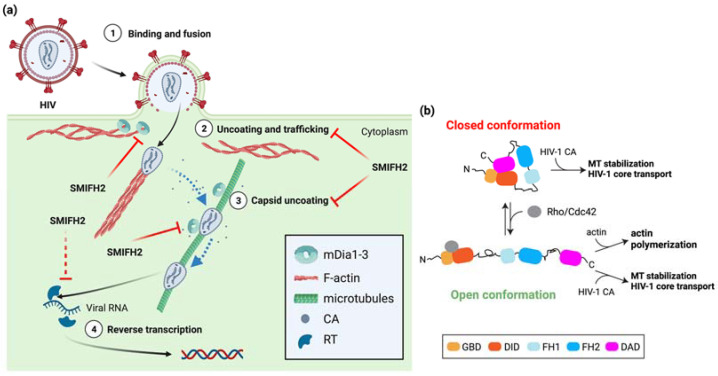
Role of mDia formins and SMIFH2 in HIV-1 entry, trafficking, and replication. (**a**) Circled numbers indicate defined HIV-1 infection stages. The involvement of mDia formins is indicated, and steps inhibited by SMIFH2 are marked by red blunt arrows. Dashed red blunt arrows highlight expected SMIFH2 effects not yet directly demonstrated in cells. Dashed light blue arrows indicate hopping of HIV-1 cores from actin filaments to MTs or their centripetal movement along MTs. Structures are not in scale, and key elements are decoded in the box. See the text for further details. Created with BioRender.com. (**b**) Possible mechanisms whereby HIV-1 cores highjack either the closed or the open conformation of mDia formins and their MT-regulatory functions. Key domains are color-coded as in Figure 1. See the text for a detailed explanation of the possible modes of action of the HIV-1 core particles in infected host cells.

**Figure 4 ijms-24-09058-f004:**
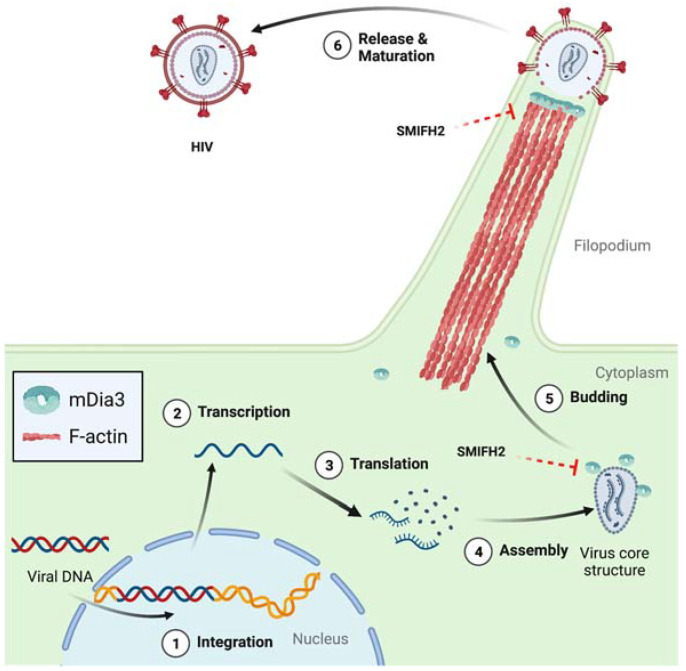
Role of mDia3 and SMIFH2 in the release of HIV-1 particles from dendritic cells. Circled numbers indicate defined HIV-1 infection stages. Dashed red blunt arrows highlight expected SMIFH2 effects not yet directly demonstrated in cells. Structures are not in scale, and key elements are decoded in the box. See text for further details. Created with BioRender.com.

**Table 1 ijms-24-09058-t001:** Novel targets on which SMIFH2 shows bioactivity. Official gene symbols of the target (its origin between brackets), PubChem bioassay identifier (AID), and source are indicated.

Target Name (Organism)	AID	Source
Chain A, RIBONUCLEASE H (HIV-1)	372	Molecular Targets Development Program
Dusp6 (*Rattus norvegicus*)	425	Burnham Center for Chemical Genomics
PTPN7 (*Homo sapiens*)	521	Burnham Center for Chemical Genomics
HSPA1A (*Homo sapiens*)	583	Burnham Center for Chemical Genomics

## Data Availability

No new data were created or analyzed in this study. Data sharing is not applicable to this article.

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
