# Peer review of "Investigating Mammalian Formins with SMIFH2 Fifteen Years in: Novel Targets and Unexpected Biology"

_ijms, 2023, doi:10.3390/ijms24109058_

Round 1

Reviewer 1 Report

The review “Investigating mammalian formins with SMIFH2 fifteen years in: novel targets and unexpected biology” is interesting and overall well-written, providing both an overview of on-target and off-target effects and insights on possible novel applications of this molecule in pathological conditions.

I have a few minor comments to help clarifying some points of the manuscript, particularly for readers that are not already familiar with the topic.

1. It might be useful to integrate (as part of Figure 1 or 2, or as a separate Table), a summary of which formins (or formin subfamily) are selectively inhibited by SMIFH2, highlighting -if known- similarity/differences in the inhibitory effect.

2. Although the possible mechanism/s by which SMIFH2 inhibits formins are described and discussed in section 5, I would probably mention before (for example in section 2) that the actual mechanism of action and binding site on formins are not known. On one hand, this makes difficult to predict off-targets effects (eg. biological targets besides formins), but on the other hand, as described, the characterization of diverse SMIFH2 biological targets may give new hints on its mechanisms of action.

3. Similarly, the fact that SMIFH2 interferes with IFN/STAT signaling by blocking the binding of IFN to its receptor is described in section 5, whereas it may be clearer to mention that already in section 3.2, to avoid confusion/misunderstanding. Currently, by reading paragraph 3.2 it was not entirely clear whether SMIFH2 acted by directly inhibiting STAT phosphorylation or by acting upstream of it, by blocking ligand (IFN)-receptor interaction.

4. In Figure 3a it is not clear which of the steps inhibited by SMIFH2 (and marked by red blunt arrows) are directly supported by literature data and which ones are a model/hypothesis about how SMIFH2 can be used to impair HIV infection not only by acting on the ribonuclease H but also by interfering with formins. Same for Figure 4. Since this is a review, and readers usually assume to find the “state of the art” on the topic, it is important to clearly define what has been tested and what is a model/hypothesis that still need a direct validation. The figure legends and/or the text can therefore be modified to clarify this point.

5. I would probably rephrase the title of the section “Discovery and characterization of SMIFH2 in vitro and in cells” as “SMIFH2 discovery and characterization in vitro and in cells”

6. please check uniformity in spelling (American English vs British English) throughout the manuscript (e.g. permeabilized or permeabilised)

Reviewer 2 Report

Review of ‘Investigating mammalian formins with SMIFH2 fifteen years in: novel targets and unexpected biology’

This is a helpful and timely review article, as the cytoskeleton field is currently re-assessing pharmacological approaches to inhibiting formins.  The compilation of information from PubChem about other SMIFH2 targets will be of particular interest to the field. I have several suggestions prior to publication.

Major suggestions/comments:

There are a few places in the manuscript where the author spoke of off- and on- target effects in a way that confused me. The first examples are in the abstract where “growing evidence of unexpected off- and on- target effects” and  “appealing off-target activities” are mentioned. The former phrase is repeated in line 97. What is the difference between an ‘unexpected on-target effect’ and an ‘appealing off-target activity’? Then later in Section 4.2 when discussing the novel SMIFH2 targets, it says “inhibition of DUSP3 and DUSP6 by SMIFH2 should be viewed as a potential off-target effect” and “if SMIFH2 inhibited HSPA1A/Hsp70-1 in cells, this would be an off-target effect”. Why are these ‘off-target’ effects but inhibition of RNase H, myosin, and IFN are not? Finally, this comes up again in lines 580-581, where it says “Unexpected potential on-target effects linking mDia2/DIAPH3 and FMN formins with p53 regulation have also come to light.” I am not clear what ‘unexpected potential on-target effects’ means here. Perhaps the author could clarify their definitions of the terms ‘on-target’ and ‘off-target’ in the introduction. 

Line 96: “a Google scholar search for ‘SMIFH2’ returned 802 entries as of March 2023”. Based on extensive literature search, our group estimates that less than half of those entries are peer-reviewed experimental studies in which SMIFH2 is used. I am happy to send the author our list of DOIs for those papers.  It was up-to-date as of a few months ago.

Line 218- 220: “Firstly, the concentration of SMIFH2 sufficient to stop non-muscle myosin 2A-dependent processes in cells is far below that needed to achieve inhibition of its activity in vitro [30].” References (and ideally concentrations) should be included for the first half of the sentence.  How different are the SMIFH2 concentrations needed to ‘stop non-muscle myosin 2A-dependent processes in cells’ and those needed to inhibit NMM2A in vitro? Furthermore, I think there is also a similar conundrum with formin inhibition studies, particularly in neurons in Qu et al (doi.org/10.1083/jcb.201701045), where 10 nM SMIFH2 is sufficient to recapitulate the effects on MT dynamics of knocking down mDia1. This is 1000-fold lower than the in vitro IC50 for formins. 

Section 4.1: The number of compounds that had activity against RNaseH (770 out of 10,000) in PubChem BioAssay 372 seems quite high. Could some context be added to help understand this number and how many also had a ‘score of 98’? It is concerning to see a hit rate that is so high and it could mean that this screening assay is prone to false positives. Related to this, how was the ‘high nanomolar-to-low micromolar IC50’ calculated?

Line 525-527: “Surprisingly, a recent study reported that the same substitution (compound 6l) had no significant effects on formins [32].” The data from that paper show a dramatic effect on the cytoskeleton when treating cells with their compound 6l, which those authors interpret as compound 6l being a formin inhibitor (an interpretation that I think deserves some scrutiny given other data that the author notes in previous sentences). The author’s wording of ‘no significant effects on formins’ might lead the reader to think that the study concluded that 6l does not inhibit formins. A suggested re-wording for this sentence: “Surprisingly, a recent study reported that the same substitution (compound 6l) did not significantly impact inhibitory activity toward formins [32].”

The discussion of covalent inhibition in section 5.2 should perhaps be more agnostic on whether this inhibition is ‘reversible’ or ‘irreversible’. As mentioned in references 29 and 32, the type of Michael addition chemistry predicted for an alkylidene thiobarbiturate reaction with a nucleophile could be readily reversible under many conditions. By the framework laid out by Houk and coworkers (doi.org/10.1021/acs.joc.6b02188), SMFH2 would be predicted to undergo reversible Michael addition. This possibility is mentioned by the authors of reference 32: “Identifying adducts formed upon reaction of IFNγ with 5k by mass spectrometry was proven challenging and may reflect the reversible nature of 1,4-additions, involving Michael acceptors…” Related to this, I think the statement on lines 553-554 “These analogues formed covalent adducts with the nucleophile groups on IFNγ, including lysines and serines…” may be too strong. In reference 32, the attachment sites could not be determined due to the aforementioned challenges with mass spectrometry. They show detection of a cross-linked protein/inhibitor complex on a gel using click chemistry (their figure 3f) and chemical shift changes in the presence of protein by fluorine NMR (their figure 3g). Separately, they show reactivity toward free lysine but *not* free serine (their figure S5). By my reading of the paper, they do not map the attachment reactive sites on IFN that react with their SMIFH2 analogs. 

Minor suggestions:

Line 34: should “humans and mammals” just be “mammals”?

The structure in Figure 2 is shown as the E isomer. I suggest making the bond to the furan ring wavy to indicate that it can be in either isomer.

Line 133: is the reference correct? 

Line 150: I think ‘…events rare cells exposed to…’ should read ‘…events rare in cells exposed to…’ 

Section 3.2: the abbreviation for interferon-gamma switches between ‘INF' and ‘IFN’. I believe that ‘IFN’ is the more common abbreviation for this protein. (This typo is also in Section 5.2)

Line 316-317: is it possible to add parentheses to make the sentence easier to read: “Knockdown of DIAPH1 or DIAPH2 strongly reduced HIV-1 entry (an effect mimicked by SMIFH2), HIV-1-induced MT stabilization, and HIV-1 reverse transcription during early stages of infection [47].” 

Line 532, ‘furane' should be ‘furan ring’

Line 535, “which do no longer allow …” should be “which no longer allow … “

see minor suggestions above
